# Asthma Mortality and Hospitalizations in Mexico from 2010 to 2018: Retrospective Epidemiologic Profile

**DOI:** 10.3390/ijerph17145071

**Published:** 2020-07-14

**Authors:** Genny Carrillo, Nina Mendez-Domínguez, Rudradeep Datta-Banik, Fernando Figueroa-Lopez, Brandon Estrella-Chan, Alberto Alvarez-Baeza, Norma Garza

**Affiliations:** 1Department of Environmental and Occupational Health, School of Public Health, Texas A&M University, 212 Adriance Lab Road, College Station, TX 77843, USA; 2Department of Health Sciences, School of Medicine, Universidad Marista, Periférico Norte Tablaje Catastral 13941, Mérida 97300, Mexico; nmendez@marista.edu.mx (N.M.-D.); rdattab1714029@a.marista.edu.mx (R.D.-B.); gfigueroa1614063@a.marista.edu.mx (F.F.-L.); bestrella1714113@a.marista.edu.mx (B.E.-C.); abaeza1094@gmail.com (A.A.-B.); 3University Health System, 4502 Medical Dr. MS 96-1, San Antonio, TX 78229, USA; norma.garza@uhs-sa.com

**Keywords:** Asthma, hospitalizations, mortality, Mexico

## Abstract

Acute respiratory infections have been established as the principal cause of disease in the Mexican population from 2000 to 2018; however, even when these diseases may aggravate asthma, there is a lack of epidemiologic evidence on the health outcomes when both conditions coexist. Learning about the asthma hospitalizations trends will help us identify monthly variation of hospitalizations, vulnerable groups, needed services, and improvements in therapeutics and prevention. This study aims to analyze the variation in asthma hospitalizations and mortality during the 2010–2018 period in Mexico. Data were obtained from the General Board of Health Information (DGIS) Open Access datasets, which were analyzed taking hospital discharges and hospital deaths recorded from 2010 to 2018 from all public hospitals nationwide. The binomial logistic regression analyses were performed to determine the association between patient ages, hospitalization month, and mortality. The death rate from asthma in Mexico decreased between 2010 and 2018. Still, the hospital mortality rate shows recent improvement; however, prognosis of hospitalized patients depends on their age, accurate diagnosis, length of hospital stay and occurrence of nosocomial infection.

## 1. Introduction

Asthma is a chronic bronchial disorder associated with obstruction of the airways, characterized by recurrent episodes of paroxysmal dyspnea, and derived from a spasmodic contraction of the bronchi. Wheezing, chest tightness, and cough are some classic symptoms of the disease due to airway hyperresponsiveness, but when these symptoms aggravate, dyspnea may become fatal if not properly controlled [1,2]. According to global research in 2014, approximately 300 million people, or 4.3% of the world population have asthma [3].

Due to differences regarding diagnostic protocols access to medical care, it is difficult to establish a global prevalence of asthma. However, one of the most important studies that have provided an overview of the disease is the International Study on Asthma and Allergies in Childhood (ISAAC) [4,5]. ISAAC showed that countries with the highest prevalence of asthma in children aged six and seven are Australia, Costa Rica, and New Zealand (26.5 to 27.1%). The countries with the highest prevalence in adolescents aged 13 and 14 years were Australia, New Zealand, Oman, Peru, Singapore, and the United Kingdom (20.7 to 28.2%) [6,7]. In Mexico, epidemiological data indicates that the prevalence of asthma diagnosed by a physician in adults is 3.3% in men and 6.2% in women [8]; however, this changes in children and adolescents, since in a study by Morales-Romero and colleagues the reported prevalence is to be 12.7% [9]. In a study in Mexico City using ISAAC methodology, the prevalence of a history of asthmatic symptoms is 19.2% in adolescents and 17% in children, emphasizing that these changes when a history of wheezing is reported in the last 12 months, the prevalence being higher in children (9.9%) in contrast to adolescents (6.8%) [10].

Even when frequent and ubiquitous, causes of death due to asthma should be a sporadic phenomenon because they are preventable with an accurate, timely diagnosis and an appropriate treatment [11]. A study conducted on 49 patients with uncontrolled asthma in Denmark reported that they complained of chest pain, dyspnea, seizures, general malaise, syncope, and palpitations before their death [12], acknowledging the importance of asthma education in patients and their relatives to identify those key symptoms, which is essential to reduce mortality [13]. 

Pelkonen et al., 2018, reported that hospitalized asthmatic patients without airway flow limitations did not have higher mortality than those with chronic obstructive pulmonary disease (*p* = 0.001) [14]. The UK National Review of Asthma Deaths (NRAD) studied 900 deaths of people of all ages in 2012 under ICD-10 code J459. The causes of death from asthma were mistakes in the first and second levels of medical care, ranging from misdiagnosis of asthma, failure to identify risk factors, and failing to provide an acceptable quality of care for acute and chronic asthma [15]. In a study by Salas and colleagues in 1994, using data from the National Statistics Institute in Mexico, a higher mortality is reported in adults over 50 years old and in children under four years old, with a hospitalization rate of 140 per 100,000 inhabitants of the country [16]. From 2000 to 2018, asthma was among the 20 leading causes of disease nationwide and the most prevalent atopic disease in the population, as reported in the morbidity yearbooks. Acute respiratory infections have been established as the principal cause of disease in the Mexican population from 2000 to 2018 [17]. Still, even when these diseases may aggravate asthma, there is a lack of epidemiologic evidence on the health outcomes when both conditions coexist [18,19]. In Mexico, the health care system consists of public or private healthcare. Private healthcare provides medical attention to patients who pay a fee for their insurance or pay for every medical service. In public healthcare, public institutions run by the government provide medical services, and patients are eligible for affiliation or insurance numbers depending on the head of the family’s employment status. Health services for unaffiliated individuals are provided by the Ministry of Health, through “Seguro Popular” [20]. The healthcare from this institution is financed from Mexican population taxes [21] and mainly benefits the lowest-income population. For statistical purposes, all public health institutions and hospitals are required to provide accurate and precise information on the medical attention provided. Each department of statistics of each hospital generates periodic information, with anonymized data from each patient in compliance with the International Classification of Diseases coding and the General Board of Health Information (DGIS) guidelines. DGIS receives and verifies each registry, applies quality control measures, and finally validates the information. Once validated, coded, and tagged with Open Access license, health information is provided thru the official webpages and made available as datasets [22,23,24]. Public health institutions in Mexico divide age groups into gross categories of pediatric, adult, and geriatric to establish the hospital department where a patient can be assisted. 

Analyzing the mortality and hospitalization trends in Mexico and its 32 states may help identify a monthly variation of hospitalizations, vulnerable age groups, and the estimated hospital services needed to improve their therapeutic approach and prevention continuously. We hypothesize that the asthma mortality rate during our study period is related to socioeconomic disparities and the accessibility to health services and hospitalization in Mexico at the time of death.

This study aims to analyze the variation in hospitalizations and asthma mortality during the 2010–2018 period in Mexico.

## 2. Materials y Methods

### 2.1. Study Design and Data Sources

In this retrospective cross-sectional epidemiological study, data were obtained from the DGIS Open Access datasets, including hospital discharges and hospital deaths recorded from 2010 to 2018 from all public hospitals belonging to the Secretaría de Salud (SSA) throughout the country. The objective of the DGIS is to identify the factors related to health and, through its use, generate a health system that meets the needs of the Mexican population. In a structured and systematized way, this national system integrates the basic information captured from hospital discharges, including the sociodemographic and hospital characteristics of the patient. In addition, procedures carried out and the deaths registered from the death certificates, stating the occurrence of death and the clinical and sociodemographic circumstances that accompanied the event, are also captured [23,24]. For the hospitalizations data, we selected the records of patients whose primary diagnosis for hospitalization was asthma, based on the International Codification of Diseases, tenth revision (ICD-10) J450 (predominantly allergic asthma), J451, J459 (non-allergic, non-specified asthma), J458 (mixed asthma), and J46X (asthmatic status). 

### 2.2. Sociodemographic Variables

Gender was assessed as binary; state of residence was categorized in 32 units, where each represents a state in Mexico. Age was described as a numeric, discrete variable and a categorical variable indicating age groups of <15 (being the cutoff point of pediatric age in hospitalized patients in Mexico), 15–65 and >65 years. Urban versus a rural place of residence was established with a cutoff point of 50,000 inhabitants. Public medical insurance, educational level, employment/occupation (if the individual was employed before his death), and marital status were available for patients aged >65 from general mortality registries.

### 2.3. Clinical Variables

The type of asthma was categorized as polytomous in allergic, non-allergic/non-specific, mixed, and status asthmaticus. The diagnosis is established in the studied hospitals in accordance to the national Clinical Practice Guidelines, which indicate signs and symptoms. In 2008, the inclusion of spirometry was highly recommended for determining an accurate asthma diagnosis. In 2017, during the guidelines’ actualization, a recommendation for a high-resolution thorax tomography for atypical and/or severe exacerbations was added [25].

A dichotomous variable indicating if the diagnosis at admission was accurate is also included. Previous hospitalizations with an asthma-related primary diagnosis as well as the length of stay were considered. The year and month of admission and discharge were also evaluated. The condition at discharge indicating if the patient was cured, improved, or expired was included in all hospitalization cases. Information of patients referred from other hospitals, emergency rooms, or ambulatory medical consultations was considered as a variable. The clinical department that discharged the patient was regarded as a categorical variable (e.g., internal medicine, pneumology department). All in-hospital infections were included as a dichotomous variable.

## 3. Statistical Analysis

For general mortality analysis, the place of occurrence and antecedent of medical assistance at the time of death were categorized and included. Asthma hospitalization rates were estimated by state per 10,000 inhabitants; general asthma mortality rates were obtained and adjusted by the state population per 100,000 inhabitants. The in-hospital mortality was calculated per 10,000 hospitalizations. For population estimates, data were obtained from the National Population Council open-access datasets.

The descriptive statistics, including the totals, proportions, and frequencies, were obtained from the categorical and numerical variables. In addition, the central tendency and dispersion measures were obtained from the numerical variables. Statistical significance was assessed through statistical hypothesis tests, by comparing proportion for dichotomous variables and mean-comparison tests for numerical data. Subsequently, logistic regression modeling was performed when dependent variables were binary or multinomial and reported as odds ratios or relative risk ratios. For multinomial variables, the reference category was the most frequent one. All the statistical analyses were conducted using the program Stata 14^®^, values with *p* < 0.05 were considered statistically significant.

## 4. Results

Between 2010 and 2018, asthma was the primary cause of hospitalizations in a total of 87,726 patients (53.2% female) admitted at public hospitals, and it was the primary cause of death in 12,495 people (53.11% female), Figure 1. The in-hospital mortality ranged between 0.82 and 17.87 per 1000 asthma-related hospitalizations, but even when general asthma mortality declined, the specific in-hospital mortality showed a significant increase. Table 1 shows the relative risk ratios (RRR) for mortality in the studied period.

### 4.1. General Mortality

Deaths due to asthma nationwide declined 26.85% in the studied period. Among the population aged <15 and >65, the reduction was 58% and 65%, respectively; however, for patients aged 15–65, the decline was not constant or significant. Table 1 shows the overall mortality rate at 10.41 per 100,000 inhabitants, with an average annual rate of 1.16 and a gender difference by female age group, with 44.9% and 53% among <15 years and >15 years, respectively. 

The percentage of asthma mortality rate by state in an out-of-hospital environment ranged between 11.08% in Puebla and 64% in Baja California, with a median of 28.75%. People who did not receive medical attention, ranged from 4.21% in Mexico City and 46.62% in Chiapas, with a median of 21.53%. The adult population who concluded elementary education (9 years) was 36%, 64% did not complete it or were illiterate. Those married at the time of death comprised 38%, and 42% did not have a usual job or were self-employed, artisans or homemakers, while 61% had an affiliation to public health insurance. 

Concerning the spatial and chronologic distribution of deaths, the peak bimester was January-February, with 30% more deaths than the expected average. Even when the National Institute of Statistics and Geography (INEGI) estimates that approximately 78% of the Mexican population lives in urban areas, asthma-related deaths in urban areas represented 54.29% of all deaths. The most frequent diagnosis in asthma-related deaths was the non-allergic, unspecific type of asthma, reported in 80.88% of cases, followed by status asthmaticus with 14.78% and allergic or mixed asthma 4.34%. Age differed significantly between the three types of asthma categories. The average age among allergic and mixed types was 17.16±1.10, status asthmaticus deaths occurred among people aged 52.25 ± 0.06, and those who died due to non-allergic unspecified asthma were aged 67.83 ± 0.22. The distribution of the specific diagnosis by age group is presented in Figure 2. When comparing the patients’ characteristics among those who died in-hospital or out-of-hospital, we used a binary logistic regression model (Table 2). The table shows that men were less prone to die in hospitals, all youngsters aged <15 died in a hospital environment, but the odds of in-hospital attention at the time of death for patients aged >65 were lower. Employment or occupational status did not differ between groups. Still, we observed that patients receiving medical attention at the time of death were commonly married, more prone to have at least elementary education, medical insurance, and living in an urban environment. 

### 4.2. Hospitalizations with Asthma as Primary Diagnosis

A total of 78,277 (87.47%) hospitalizations corresponded to a first-time admission during November-December, and most hospitalizations were associated with pediatric patients, with 72.53% <15 years, 24.01% between 15 to 65 years, and 3.71% were aged over 65. Table 3 shows patients’ characteristics emphasizing on hospitalizations, and outcomes depending on their type of asthma. The younger average age is related to allergic asthma, which was also more frequent among male patients, less commonly diagnosed accurately at admission or having a previous hospitalization. In-hospital infection occurred in 307 cases (0.34%), and it was directly correlated with more extended hospital stay (5.79 versus 3.06 days, *p* < 0.001) when comparing the length of stay (LOS) of infected versus non-infected patients. On the other hand, infection increased the odds of death among infected patients compared to non-infected (OR = 10.93:1 *p* < 0.001). Figure 3 illustrates increased mortality at the end of the studied period. The number of patients who died in Mexico hospitals between 2010 to 2018 was 379, with a mortality rate of 4.32 per 1000 hospitalizations.

The in-hospital odds for mortality varied in association with the epidemiological and clinical features of the patients. Table 4 shows the association between epidemiological and clinical manifestations of patients concerning fatal outcomes. Hospital stay increased the odds of death; pediatric patients were less prone to undercome deadly results, and male patients showed higher death odds. The fatal type of asthma was the mixed type, and allergic asthma showed higher odds of recovery. An accurate initial diagnosis demonstrated to be a protective factor. A first-time hospitalization and in-hospital infections correlated with higher odds of mortality.

## 5. Discussion

In the present study, we have provided an overview of asthma-related mortality and its clinical course outcomes of hospitalizations at public institutions in Mexico between 2010 and 2018. We found that during the studied period, the mortality rate in Mexico has decreased continuously. However, the diagnosis is influenced by certain sociodemographic factors such as medical assistance at the time of death. Among hospitalized patients, epidemiological characteristics are related to clinical courses and outcomes among patients. Consistent with previous studies in the USA, Brazil, France, Kuwait, Costa Rica, and Galicia, we found a reduced asthma hospitalization rate for all age groups throughout the studied period [26,27,28,29,30,31,32,33]. Vargas et al. reported a similar decreased health services trend of asthma over the years translating, into a decrease in hospitalizations [30]. Another finding in this study was that most deaths from asthma in Mexico, 2010–2018, occurred in January–February, which is similar to the results González and colleagues reported from Galicia (in the period 1993–2007), which occurred from January to March [34].

A study done in Portugal reported that during 2000–2010, their hospitalizations occurred in December–January [35]. In Mexico, we found that most asthma admissions occurred in the months of November–December; however, a study done by Silva et al. in Brazil from 2001–2007 reported that the majority of hospitalizations for asthma occurred in March and May [26]. These discrepancies may be due to differences in seasonality between geographical locations across the equator. In the present study, we found that the hospital mortality rate in Mexico (during 2010–2017) was higher for mixed asthma than for other asthma types, which is different from a study in England during 2000–2005. In such a study, asthmatic status was the type of asthma associated with the highest hospital mortality rate [36]. Other factors associated with in-hospital death were related to the male gender. Our results agree with Chang et al. reported that in-hospital infection (pneumonia, genitourinary disease, and septicemia) were independent risk factors of death in patients admitted for asthma exacerbation. [37].

Chua and colleagues found that the hospitalization rate in patients aged 0–14 years for asthma decreased in a uniform rate in Australia and Singapore from 2010 to 2018 in patients of the same age; in contrast to that study, in Mexico, the period from 2014 to 2016 shows that the mortality rate in 0–15 years old, was on the rise [38]. However, in the present study, the Mexican population showed that the hospitalization rate in pediatric patients (0–15 years) was higher than in other age groups and that socioeconomic disparities exist when related to mortality [26,38]. Our results agree with similar studies, with higher frequency and mortality rate from asthma in women and elderly patients; only allergic-type asthma was more frequent among males and younger people [28,31]. García et al. reported similar results in the adult population in a study carried out in Mexico City, where the prevalence of asthma in women is higher, and have a more significant association of having been admitted to hospital for any respiratory disease (OR = 51.5:1 *p* < 0.001) [8]. One of the main results of the present study is that for patients 0–15 years old, the hospitalization rate for asthma in Mexico (2010–2018) was higher in males, but it was higher in females for the other age groups. These results are similar to studies carried out in Kuwait by Ziyab et al. and Santos et al. carried out in Portugal [29,35]. However, in a study carried out in Mexico City by Del-Rio-Navarro and colleagues, the difference in the prevalence of the disease by gender is not significant in children and adolescents in the general population. Another outcome found in the study period was that the frequency of mortality from asthma in Mexico was higher in people without occupation than to those who had it [10]. Gupta et al. in England had a similar finding with a higher frequency of mortality in those with low socioeconomic status, consistent with our results from Mexico [39].

### 5.1. Limitations

As any other cross-sectional design, the present study has particular limitations: first are those that derive from data obtained from the General Board of Health Information (DGIS) Open Access datasets from all public hospitals nationwide. Therefore, this study does not include private hospitals where clinical epidemiology of asthma may significantly differ; nevertheless, the information from general mortality was obtained from public and private health institutions and out of hospital mortality. The study does not allow us to analyze patient behaviors that might have contributed to increased mortality, and it is limited due to the timeframe of the data obtained.

### 5.2. Public Health and Policy Implications

The findings of this study have important public health and policy implications. This study suggests that the death rate decreased from 2010 to 2018, and hospital mortality has decreased. An accurate diagnosis is critical to provide patients the adequate treatment to decrease the LOS in the hospital and the number of visits to the Emergency Department. Given that asthma education is an effective way to improve knowledge for asthma management and health outcomes of patients with asthma, it would be important to offer asthma education to patients and their families. Asthma education normally focuses on signs and symptoms. However, increasing knowledge of asthma triggers inside and outside their homes, and adequate management of their medication adherence will prevent asthma attacks and overutilization of healthcare resources and death [39,40,41,42]. 

This study will also help hospital leaders and physicians who serve asthma patients to understand the mortality and hospitalizations trends and the importance of offering an accurate diagnosis complemented with patient education. 

## 6. Conclusions

The death rate from asthma in Mexico decreased from 2010 to 2018. However, the hospital mortality rate shows a recent rebound at 5.48 per 1000 asthma-related hospitalizations in Mexico in 2018, showing an increase in relation to 2017, with a rate of 3.7. At the same time, the prognosis of hospitalized patients depends on each patient’s epidemiological aspects, such as age, and clinicians’ accurate diagnosis, LOS, and occurrence of nosocomial infection. The present study can help direct future research lines in which asthma is addressed at a more specific level that includes the different clinical presentations and various types of asthma in different age groups of hospitalized and deceased patients.

## Figures and Tables

**Figure 1 ijerph-17-05071-f001:**
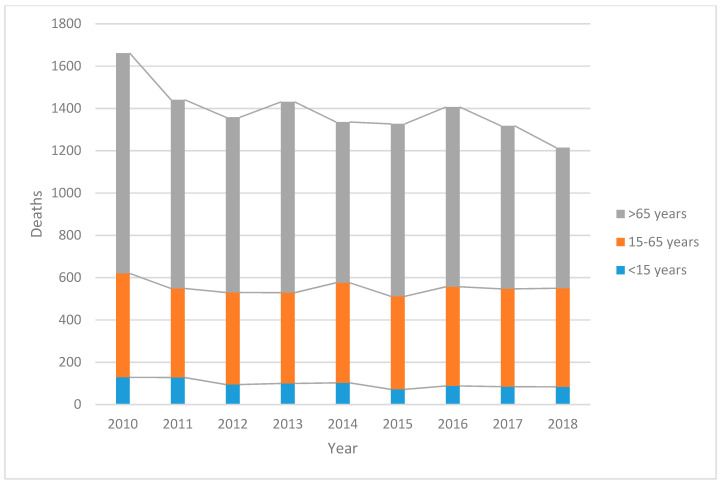
Number of deaths due to asthma in Mexico from 2010 to 2018 (*n* = 87,726).

**Figure 2 ijerph-17-05071-f002:**
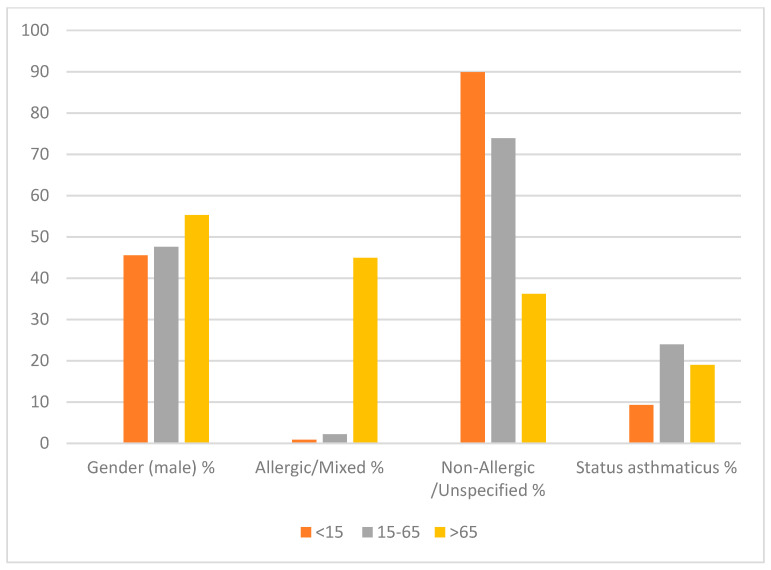
Distribution of mortality due to asthma in Mexico between 2010 and 2018 by diagnosis and age group (*n* = 12,483).

**Figure 3 ijerph-17-05071-f003:**
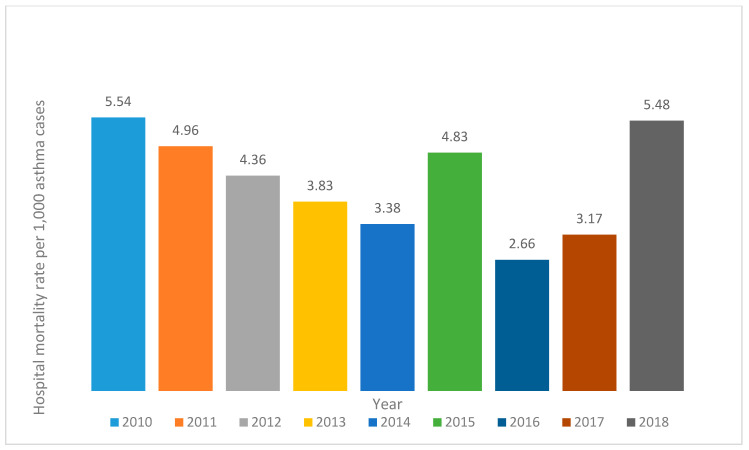
Asthma mortality rate per 1000 hospitalizations in Mexico from 2010 to 2018.

**Table 1 ijerph-17-05071-t001:** Relative risk ratio of death in hospital environment due to asthma in Mexico between 2010 and 2018.

Year	Relative Risk Ratio	Standard	*z*	*p*	Confidence Intervals 95%	
		Error			Lower	Upper
2010	*Base outcome RRR = 1.00*					
2011	0.99	0.08	−0.18	0.856	0.84	1.16
2012	1.19	0.10	1.99	*0.046*	1.000	1.41
2013	1.23	0.11	2.4	*0.017*	1.04	1.45
2014	1.12	0.10	1.31	0.189	0.95	1.33
2015	1.01	0.09	0.16	0.874	0.86	1.2
2016	1.21	0.10	2.24	*0.025*	1.02	1.44
2017	1.28	0.11	2.79	*0.005*	1.08	1.52
2018	1.36	0.13	3.39	*0.001*	1.14	1.63

RRR = Relative Risk Ratio.

**Table 2 ijerph-17-05071-t002:** Association between sociodemographic characteristics and the odds of receiving in-hospital attention at the time of death in Mexico between 2010–2018 (*n* = 12,483).

Characteristic	Odds Ratio	Standard	*z*	*p*	Confidence Intervals 95%	
		Error			Lower	Upper
Gender (male)	0.70	0.04	−7.06	0.000	0.63	0.77
Age <15	1.46	0.11	4.82	0.000	1.25	1.70
Age 15–65	2.91	0.13	24.16	0.000	2.67	3.18
Age >65	0.39	0.02	−17.84	0.000	0.35	0.43
Employment	0.94	0.05	−1.15	0.250	0.85	1.04
≥Elementary school	1.18	0.08	2.66	0.008	1.05	1.34
Married	1.59	0.08	9.00	0.000	1.44	1.76
Medical Insurance	2.57	0.14	16.68	0.000	2.30	2.87
Urban residence	3.43	0.18	23.77	0.000	3.09	3.79

R^2^ = 17.79.

**Table 3 ijerph-17-05071-t003:** Characteristics of patients hospitalized in Mexico with asthma as primary diagnosis in 87,726 cases occurred between 2010 and 2018 (showing means/percentages followed by standard deviations.

Type of Asthma	Age (Years)		Length of Stay (Days)		Male		Medical Insurance		From Emergency Room		From Ambulatory Consultation		Accurate Initial Diagnosis		First Time Hospitalized		In-Hospital Infection		Mortality Per 1000 Patients	
	*n*	SD	*n*	SD	*n*	SD	*n*	SD	*n*	SD	*n*	SD	*n*	SD	*n*	SD	*n*	SD	*n*	SD
Allergic	5.40	0.12	2.69	0.04	57.23	0.57	72.75	0.52	96.37	0.22	1.75	0.15	94.52	0.26	93.89	0.28	0.19	0.05	1.85	0.30
Non-Allergic	16.84	0.10	3.08	0.02	47.56	0.24	68.07	0.22	93.56	0.12	4.30	0.10	95.27	0.10	90.41	0.14	0.33	0.03	4.52	0.27
Mixed	18.09	1.11	2.86	0.12	47.37	2.45	72.49	2.19	90.67	1.42	5.26	1.09	96.17	0.94	88.76	1.55	0.72	0.41	9.30	0.63
Status asthmaticus	15.01	0.10	3.13	0.02	48.80	0.26	66.99	0.25	94.27	0.12	4.20	0.11	96.64	0.09	82.54	0.20	0.39	0.03	4.29	0.27

**Table 4 ijerph-17-05071-t004:** Epidemiologic and Clinical characteristics associated to fatal outcome among patients hospitalized due to asthma in Mexico from 2010 to 2018 (*n* = 87,726).

Death	Odds Ratio	Std. Err.	*z*	*p*	(95% Confidence Intervals)	
Hospital Stay per age	*1.02*	*0.00*	*5.33*	*0.000*	*1.01*	*1.03*
<15 years	*0.05*	*0.03*	*−5.30*	*0.000*	*0.02*	*0.15*
15–65 years	0.65	0.35	−0.81	0.418	0.23	1.85
>65 years	2.05	1.09	1.36	0.175	0.73	5.81
Gender (male)	*1.47*	*0.20*	*2.88*	*0.004*	*1.13*	*1.91*
From Emergency room	1.92	1.13	1.11	0.269	0.60	6.07
From External consultation	0.94	0.65	−0.09	0.930	0.24	3.65
Allergic	*0.22*	*0.10*	*−3.38*	*0.001*	*0.09*	*0.53*
Mixed type asthma	*3.33*	*1.68*	*2.37*	*0.018*	*1.23*	*8.97*
Non-Allergic/Unspecific	*1.36*	*0.17*	*2.42*	*0.015*	*1.06*	*1.74*
Status asthmaticus	0.91	0.12	−0.73	0.466	0.71	1.17
Accurate initial diagnosis	*0.11*	*0.02*	*−15.34*	*0.000*	*0.09*	*0.15*
First time Hospitalized	*1.94*	*0.43*	*3.02*	*0.003*	*1.26*	*2.98*
In-Hospital Infection	*9.26*	*3.46*	*5.96*	*0.000*	*4.45*	*19.25*

Pseudo R^2^ = 0.2029.

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
