# Peer review of "Asthma Mortality and Hospitalizations in Mexico from 2010 to 2018: Retrospective Epidemiologic Profile"

_ijerph, 2020, doi:10.3390/ijerph17145071_

Round 1

Reviewer 1 Report

The study's objective is to analyze variation in hospitalizations and asthma mortality during the 2010-2018 period in 96 Mexico. The objective is too broad. Analyses as such can not be accepted. The main hypotheses should be identified.

A short description of the health system in Mexico would be helpful.

Methods – remarks. It is stated that: “for population estimates, data were obtained from the National Population Council open-access datasets”. The completeness and the quality of information have not been addressed.

Material is unique and worth investigation. it includes a  total of 87,726 patients admitted at public hospitals and it was the primary cause of death in 12,495 people. The most important is finding of a wide range of  in-hospital mortality  statistics - 0.82 - 17.87 per 1,000 asthma-related hospitalizations,

Table 2 - the comparison groups for the age – are not identified. On the other hand – the occupation category is not clear.

Discussion- the selection of the reviewed publication should be tuned to the selected hypothesis.

Author Response

Dear reviewers,

We are thankful for the time and dedication you invested in critically reviewing our manuscript.

We made a thoughtful revision of the manuscript and corrected following your observations and comments to incorporate our manuscript changes. Please find in blue font the specific responses to your valuable feedback.

Reviewer 1

Comments and Suggestions for Authors

  • The study's objective is to analyze variation in hospitalizations and asthma mortality during the 2010-2018 period in Mexico. The objective is too broad. Analyses, as such, cannot be accepted. The main hypotheses should be identified.

Response: Thank you for your comment, we have corrected, identified and added the hypothesis as follows: We hypothesize that the asthma mortality rate during our study period is related to socioeconomic disparities and the accessibility to health services and hospitalization in Mexico at the time of death.

  • A short description of the health system in Mexico would be helpful.

Response: We are thankful; it surely will enrichen the manuscript and ad clarity for our readers. We have added a paragraph regarding the health system in Mexico: 

In Mexico, the health care system consists of public or private healthcare. Private healthcare provides medical attention to patients who pay a fee for their insurance or pay for every medical service when needed. In public healthcare, public institutions run by the government provide medical services, and patients are eligible for affiliation or insurance numbers depending on the employment status of the head of the family. Health services for unaffiliated individuals are provided by the Ministry of Health, through "Seguro Popular" [20]. The healthcare from this institution is financed from Mexican population taxes [21] and mainly benefits the lowest-income population. 

  • Methods – remarks. It is stated that: "for population estimates, data were obtained from the National Population Council open-access datasets." The completeness and the quality of information have not been addressed.

Response: We added the information regarding the topic as follows:

For statistical purposes, all public health institutions and hospitals must provide accurate and precise information on the medical attention provided. Each department of statistics of each hospital generates periodic information, with anonymized data from each patient in compliance with the International Classification of Diseases coding and the General Board of Health Information (DGIS) guidelines. DGIS receives and verifies each registry, applies quality control measures, and finally validates the information. Once validated, coded, and tagged with Open Access license, health information is provided thru the official webpages and made available as datasets [22-24].

  • Material is unique and worth investigation. It includes 87,726 patients admitted to public hospitals, and it was the primary cause of death in 12,495 people. The most important is finding a wide range of in-hospital mortality statistics - 0.82 - 17.87 per 1,000 asthma-related hospitalizations.

Table 2 - the comparison groups for the age – are not identified. On the other hand, the occupation category is not clear.

Response: Thank you for your observation. We added specific diagnosis by age groups stratification in the new figure, so that age groups' specification appears in the manuscript before the table.

We have clarified the "occupation" category, as it means whether the patient was employed by death.

Discussion- the selection of the reviewed publication should be tuned to the selected hypothesis.

Response: Thank you, we have now directed our discussion to be concordant to our hypothesis.

Reviewer 2 Report

Authors wished to investigate  the variation in hospitalizations and asthma mortality during the 2010-2018 period in Mexico. The study is a retrospective cross-sectional epidemiological study, which used data obtained from the General Board of Health Information (DGIS) Open Access  datasets, where hospital discharges and hospital deaths  recorded from 2010 to 2018 from all public hospitals nationwide could be found.

They found that asthma mortality in Mexico decreased from 2010 to 2018, particularly in patients < 15 or > 65 years. Age, clinicians’ accurate diagnosis, length of hospital stay and nosocomial infection were found to be the relevant variables predicting mortality.

The study is interesting  and well written

Minor point: could Authors explain better the meaning of “clinicians’ accurate diagnosis” ?

Author Response

Reviewer 2

Comments and Suggestions for Authors

  • Minor point: could Authors explain better the meaning of “clinicians’ accurate diagnosis”?

Response: We now see that can clarify this point to our readers.

We have added the following paragraph.

The diagnosis is established in the studied hospitals in accordance to the national Clinical Practice Guidelines, which indicate signs and symptoms.  In 2008, the inclusion of spirometry was highly recommended for determining an accurate asthma diagnosis. In 2017, during the guidelines' actualization, a recommendation for a high-resolution thorax tomography for atypical and/or severe exacerbations was added [25].

Thus, clinicians´ accurate diagnosis means that when clinicians made specific and precise diagnoses, including the type of asthma, the patients had better outcomes. In contrast, an inaccurate hospitalization diagnosis (e.g., when the patients were hospitalized with unrelated diagnosis and then changed to asthma) patients had higher odds for in-hospital mortality or prolonged hospital stay.

We have rephrased discussion also to ad clarity and changed to “initial accurate diagnosis” instead of “clinicians´ accurate diagnosis.”

Round 2

Reviewer 1 Report

The authors introduced most of the changes I suggested.

I do not have any further comments